# ADAPT: Alzheimer's Diagnosis through Adaptive Profiling Transformers

## Abstract

Automated diagnosis of Alzheimer's Disease (AD) from brain imaging, such as magnetic resonance imaging (MRI), has become increasingly important and has attracted the community to contribute many deep learning methods. However, many of these methods are facing a trade-off that 3D models tend to be inefficient in training and inferencing while 2D models cannot capture the full 3D intricacies from the data. In this paper, we introduce a new model structure for diagnosing AD, and it can complete with 3D model's performances while essentially is a 2D method (thus computationally efficient). While the core idea lies in building different blocks on different views according to physicians' diagnosing perspectives, we introduce multiple components that can further benefit the model in this new perspective, including adaptively selecting the number of sclices in each dimension, and the new attention mechanism. In addition, we also introduce a morphology augmentation, which also barely introduces new computational loads, but can help improve the diagnosis performances due to its alignment to the pathology of AD. We name our method ADAPT, which stands for Alzheimer's Diagnosis through Adaptive Profiling Transformers. We test our model from a practical perspective (the testing domains do not appear in the training one): the diagnosis accuracy favors our ADAPT with 4.5% improvement, while ADAPT uses at leat 14% less parameters than the state-of-the-art models.

## 1 Introduction

Alzheimer's disease (AD) is a highly common neurodegenerative disorder that is usually diagnosed by structural alterations of the brain mass. Assessing an AD usually involves the acquisition of magnetic resonance imaging (MRI) images, since it offers accurate visualization of the anatomy and pathology of the brain Zhou et al. (2023b). To overcome the vulnerability of misdiagnosis Despotović et al. (2015) and to speed the diagnosis process, the community has been using machine intelligence to help physicians diagnose AD diseases Jo et al. (2019).

Considering the complex structure of brain magnetic resonance imaging (MRI), in recent years, Convolutional Neural Networks (CNNs) have been established with a dominant performance in the AD-related field Salehi et al. (2020); Farooq et al. (2017), due to their effectiveness in extracting meaningful spatial hierarchical features from complex images. Many methods Zhu et al. (2021); Wen et al. (2020) try to learn the characteristics of AD using CNN-based models. However, the original MRI is complex 3D data, with the proposed 3D model, the input of the 3D convolution operation introduces a third dimension, which greatly increases the burden on the computer. So they use a bag of patches selected from the skull-stripped brain region. These approaches disregard the global context information, which can have a substantial impact on accurately identifying lesions during inference Wang et al. (2022). Moreover, CNNs are not well-suited for mining global long-dependent information due to their inherent focus on extracting local information Luo et al. (2016); Dosovitskiy et al. (2020).

Transformers Vaswani et al. (2017) have also been widely used in medical imaging because of their superior performances over CNNs. Such spatial relationships are crucial in 3D MRI images for Alzheimer's diagnosis Iaccarino et al. (2021), where understanding cross-sectional interdependencies is the key. However, transformer-based methods have yet to see widespread use in 3D medical image diagnosis. A primary reason is that due to a lack of inductive bias of locality, lower layers of ViTs can

not learn the local relations well, leading to the representation being unreliable Zhu et al. (2023). Also, 3D medical images are usually complex, making ViTs hard to pay attention to a special local feature that will play a crucial role in Alzheimer's diagnosis. Moreover, in 3D medical imaging, the scarcity of datasets, largely due to ethical considerations that restrict access Setio et al. (2017); Simpson et al. (2019), costly annotations Yu et al. (2019); Wang et al. (2023), class imbalance challenges Yan et al. (2019), and the significant computational demands of processing high-dimensional data Tajbakhsh et al. (2020), is a notable issue.

At the same time, these models typically treat all the dimensions in the same way. In contrast, when physicians read the MRI, they usually pay different attentions to different dimensions of the images, according to the atrophic patterns of the brain. This adaptive strategy of the physicians allows them to diagnose more efficiently and accurately.

Inspired by the above, we propose ADAPT, a pure transformer-based model that leverages the captured different features from each view dimension more smartly and efficiently. Our goal is to classify Alzheimer's disease (AD) and normal states in 3D MRI images. ADAPT factorizes 3D MRI images into three 2D sequences of slices along axial, coronal and sagittal dimensions. Then we combine multiple 2D slices as input and use a 2D separate transformer encoder model to classify. At the same time, we also build attention encoders across slices from the same dimension and the attention encoders across three dimensions. These encoders can help to efficiently combine the feature information better than just keep training using the slices altogether. Benefiting from the special encoder blocks with morphology augmentation and adaptive training strategy, ADAPT can learn the AD pathology just using a few slices instead of inputting all 2D images, which can further reduce memory footprint. The detailed architecture is shown in Section 3. Our contributions are as follows:

- We proposed a new transformer-based architecture to solve the real-world AD diagnosis problem.

- We proposed a novel cross-attention mechanism and a novel guide patch embedding, which can gather the information between slices and sequences better.

- Considering the structure and difference between AD and normal MRI images, we designed the morphology augmentation methods to augment the data.

- We proposed an adaptive training strategy in order to guide the attention of our model, leading the model to adaptively pay more attention to the more important dimension.

- Overall, we name our method ADAPT, which is evaluated as the state-of-the-art performance among all the baselines while occupying minimum memory.

## 2 RELATED WORKS

### 2.1 3D VISION TRANSFORMER

The recent success of the transformer architecture in natural language processing Vaswani et al. (2017) has garnered significant attention in the computer vision domain. The transformer has emerged as a substitute for traditional convolution operators, owing to its capacity to capture long-range dependencies. Vision Transformer (ViT) Dosovitskiy et al. (2020) introduces transformer architecture into the computer vision field and starts a craze in combining transformers and images together. Many works have demonstrated remarkable achievements across various tasks, with several cutting-edge methods incorporating transformers for enhanced learning.

Some attention-based methods have been proposed for 3D image classification. COVID-VIT Zhou et al. (2023a) uses 3D vision transformers to exploit CT chest information for the accurate classification of COVID. I3D Carreira & Zisserman (2017) proposes a new two-stream inflated 3D ConvNet to learn seamless spatio-temporal feature extractors from video, which can be used to do human action classification. At the same time, many existing works also deal with 3D object detection problems. Pointformer Pan et al. (2021) captures and aggregates local and global features together to do both indoor and outdoor object detection. 3DERT Misra et al. (2021) proposes an encoder-decoder module that can be applied directly on the point cloud for extracting feature information, and then predicting 3D bounding boxes. Also, image segmentation is a hot topic in the both computer vision

and medical imaging fields. Swin UNETR Hatamizadeh et al. (2021) projects multi-modal input data into a 1D sequence of embedding and uses it as input to an encoder composed of a hierarchical Swin Transformer Liu et al. (2021).

**Key Differences:** These models are all using 3D architecture to deal with 3D input, which is inefficient in medical field due to the high value of medical images and limited dataset size. Unlike them, our 2D ADAPT utilizes different blocks to first extract features among different slices and dimensions, then use a cross-attention mechanism to combine these features together, which can better release the abilities of transformer architecture.

## 2.2 Deep Learning for Medical Image Analysis

With the success of deep learning models, extensive research interest has been devoted to deep learning for the development of novel medical image processing algorithms, resulting in remarkably successful deep learning-based models that effectively support disease detection and diagnosis in various medical imaging tasks Chen et al. (2022). U-Net and its variants dominate medical image analysis, which is widely used in image segmentation. Attention U-Net Oktay et al. (2018) incorporates attention gates into the U-Net architecture to learn important salient features and suppress irrelevant features.

For medical image classification, AG-CNN Guan et al. (2018) uses the attention mechanism to identify discriminative regions from the global 2D image and fuse the global and local information together to better diagnose thorax disease from chest X-rays. MedicalNet Chen et al. (2019) uses the resnet-based He et al. (2016) model with transfer learning to solve the problem of lacking datasets. DomainKnowledge4AD Zhou et al. (2023b) uses ResNet18 to extract high-dimensional features and proposes domain-knowledge encoding which can capture domain-invariant features and domain-specific features to help predict AD. ACS Yang et al. (2021) leverages large amount of 2D images and expands pretrained 2D convolutions to 3D on different view dimensions to solve 3D problems. M3T Jang & Hwang (2022) tries to leverage CNNs to capture the local features and use traditional transformer encoders for a long-range relationship in 3D MRI images.

**Key Differences:** These methods usually focus on CNN based model to extract and combine features, which has been outperformed by transformer-based models. ACS tries to deal with 3D problems on different view dimensions, nevertheless, the lack of an efficient fusion layer and the pure CNN-based architecture will lead to a terrible understanding of the spatial relationship in 3D images. M3T tries to concate transformer blocks after CNNs, however, they propose a much bigger model and treat all slices as the same which is inefficient. In our work, we use a pure transformer-based model with different kinds of encoders to do Alzheimer's classification and have demonstrated ADAPT can outperform other deep learning models in both classification accuracy results and model size.

## 3 Methodology

ADAPT mainly consists of three main parts: morphology augmentation, ADAPT encoder blocks and adaptive training strategy. As shown in Figure 1, when a 3D MRI image comes in, it will be first split into three sequences according to coronal, sagittal and axial view, then the images will be augmented to align the pathology feature of AD with morphology augmentation (section 3.3). Then the sequences will be encoded by different encoders to fully capture the features (section 3.1). Before the next iteration, adaptive rank training will rank the importance of each view with the output attention score from the final encoder, and resplit the next 3D image (section 3.4).

## 3.1 Model Architecture

In the real-world setting, while physicians diagnosis alzheimer's disease with MRI images, the physicians will pay different attentions to different views according to the brain pattern. Because clinicians usually diagnose AD using 2D slices but not the whole 3D MRI, we conjecture 2D slices may contain more valuable information. Thus the design of ADAPT is inspired by this setting. At the same time, manipulating spatial information is crucial for a variety of goals and cognitive abilities Galati et al. (2010), and clinicians may use spacial information in their brain when diagnosing the AD-related images. Thus to keep the ability of ADAPT in modeling the spatial information, ADAPT mainly consists of 4 blocks:

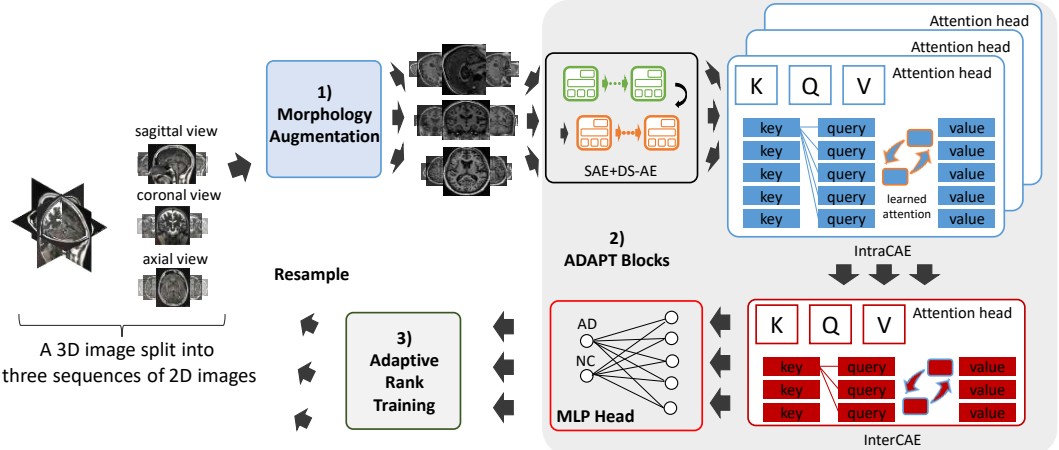

Figure 1: The detailed architecture for our ADAPT. ADAPT consists of three main modules: 1) Morphology Augmentation for atrophy expansion and reduction. 2) Four blocks: **S**elf-**A**ttention **E**ncoders (SAE) across three views, **D**imension-specific **S**elf-**A**ttention **E**ncoders (DS-AE), **Intra**-dimension **C**ross-**A**ttention **E**ncoders (IntraCAE), **Inter**-dimension **C**ross-**A**ttention **E**ncoders (InterCAE) with fusion attention mechanism. 3) Adaptive Rank Training for dimension-based attention score calculation. After ranking, the score will be used to resample different amounts of 2D images on different views.

- **S**elf-**A**ttention **E**ncoders (SAE) across three views
- **D**imension-specific **S**elf-**A**ttention **E**ncoders (DS-AE)
- **Intra**-dimension **C**ross-**A**ttention **E**ncoders (IntraCAE)
- **Inter**-dimension **C**ross-**A**ttention **E**ncoders (InterCAE)

These encoders can not only extract and fuse features from local and global patterns but also assign different attentions to different views. To be specific, first, to better obtain the complete information of the 3D image, we cut each image along three views: sagittal view (along x-axis), coronal view (along y-axis), and axial view (along z-axis). We use $n$ images from each view as the model input. Then similar to ViT, ADAPT also uses the image patch and patch embedding method to embed the 2D images into 3 sequences including $3 \times n$ slices with guide patch embedding layer $\mathbf{x}_{guide}$, then concatenates them together as the input to the transformer encoders (Eq. 1). The guide patch embedding aims to reshape the whole sequence into a sequence of flattened 2D patches that has the same shape as the sequence after the normal patch, which means the guide patch embedding has the input channel with the number $3 \times n$. With the guide patch embedding design, we can use 3D models to extract the global information and add it to each special slice sequence. Because our model mainly focuses on 2D slice dimension, guide patch embedding can help to keep the relative position information of 3D brain.

$$\mathbf{S}_0 = [\mathbf{x}_{class}; \underbrace{\mathbf{x}_{p_1} + \mathbf{x}_{guide}; \cdots ; \mathbf{x}_{p_n} + \mathbf{x}_{guide}}_{sagittal}; \underbrace{\cdots ; \mathbf{x}_{p_{2n}} + \mathbf{x}_{guide}}_{coronal}; \underbrace{\cdots ; \mathbf{x}_{p_{3n}} + \mathbf{x}_{guide}}_{axial}] \tag{1}$$

$$\mathbf{S}_0 = \mathbf{S}_0 + \mathbf{E}_{pos} \qquad \mathbf{E}_{pos} \in \mathbb{R}^{(3 \cdot n \cdot N + 1) \times D} \tag{2}$$

Second, the lower layer encoders learn the bias attention among multiple slices and multiple views. To be more specific, the shared **S**elf-**A**ttention **E**ncoders (SAE) across three view dimensions are designed to learn not only the attention of the slice itself but also the relationship between all slices. The designed encoder can realize global information extraction for the first time. These encoders can also help to keep the relative position information of 3D MRI. These networks are Siamese networks Guo et al. (2017) which share the same weights.

$$\mathbf{S}_0^s = [\mathbf{x}_{class}^s; \mathbf{x}_{p_s}] \qquad s \in (1, 3 \cdot n) \tag{3}$$

$$\mathbf{S}_l^s = \text{SAE}(\mathbf{S}_{l\text{-}1}^s) \qquad l = 1...L_{\text{SAE}} \tag{4}$$

The **D**imension-specific **S**elf-**A**ttention **E**ncoders (DS-AE) also aim to learn the attention of the slice itself. However, compared with SAE, these encoders focus more on the relationship between the slices from the same dimension sequence. These encoders can better extract the local features from the same view dimension. This will fill the gap that transformers cannot capture the local features well however the local embeddings of different brain tissues (such as hippocampus and cortex) are really important in AD diagnosis. In the following equation, t means the three different views.

$$\mathbf{S}_l^{t \cdot s} = \mathrm{DSAE_t}(\mathbf{S}_{l\text{-}1}^{t \cdot s}) \qquad s \in (1, n), t \in (1, 3), l = (L_{\mathrm{SAE}} + 1)...(L_{\mathrm{SAE}} + L_{\mathrm{DSAE}}) \tag{5}$$

We will fusion the local features from the same dimension first. So we design **Intra**-dimension **C**ross-**A**ttention **E**ncoders (IntraCAE). Here ADAPT will apply cross embedding mechanism to the input embeddings. (Details are in section 3.2.) After the IntraCAE, the embeddings will gather the features from different slices of the same view sufficiently.

$$\begin{aligned} \mathbf{S}_l^{t \cdot s} =& \mathrm{IntraCAE_t}(\mathbf{S}_{l\text{-}1}^{t \cdot s}) \qquad s \in (1, n), t \in (1, 3), \\ & l = (L_{\mathrm{SAE}} + L_{\mathrm{DSAE}} + 1)...(L_{\mathrm{SAE}} + L_{\mathrm{DSAE}} + L_{\mathrm{IntraCAE}}) \end{aligned} \tag{6}$$

After combining the features between slices of the same dimension independently, the last **In-ter**-dimension **C**ross-**A**ttention **E**ncoders (InterCAE) are proposed to learn the inter-dimension relationship among different sequences from different views. This is corresponding to the SAE layer and will gather the global features together. InterCAE will apply cross embedding mechanism again into the view-dependent embeddings.

$$\begin{aligned} \mathbf{S}_l^{t} =& \mathrm{InterCAE_t}(\mathbf{S}_{l\text{-}1}^{t}) \qquad t \in (1, 3) \\ & l = (L_{\mathrm{SAE}} + L_{\mathrm{DSAE}} + L_{\mathrm{IntraCAE}} + 1)...(L_{\mathrm{SAE}} + L_{\mathrm{DSAE}} + L_{\mathrm{IntraCAE}} + L_{\mathrm{InterCAE}}) \end{aligned} \tag{7}$$

Finally, the $[class]$ tokens of the output from three dimensions will be averaged and sent to Layer Norm and classification MLP head to get the final diagnosis result: AD or normal.

## 3.2 Fusion Attention Mechanism

The above architecture will allow us to learn the intricies of AD pathologies along three different dimensions. However, the complicatedness of AD will require the model to thoroughly integrate the information from these three dimensions. Thus, we propose a cross-attention mechanism, namely fusion attention. The fusion attention adds the embeddings together directly. However, different from simply adding them together one by one, it adds the embeddings representing the patches but not the tokens. Note that the $[class]$ token of each embedding has aggregated the information from one slice in previous encoders, so this operation will let the embeddings more focus on themselves when learning attention. At the same time, it can also extract the feature information from other slices or dimensions. The fusion attention applied to both IntraCAE and InterCAE, but here we use IntraCAE as an example:

$$\mathbf{S}_l^{t \cdot s} = \mathbf{x}_{class}^{t \cdot s} \oplus (\mathbf{x}_{p_{(t-1) \cdot n+1}} + \cdots + \mathbf{x}_{p_{t \cdot n}}) \quad \text{where} \quad s \in (1, n), t \in (1, 3) \tag{8}$$

In a more formal way, the traditional attention mechanism is shown as Eq. 9. After fusing these two embeddings, the $K$ matrix of the first embedding will consist of the $K$ value corresponding to the $[class]$ token from the first embedding, and the $K$ matrix corresponding to fusion embedding, similarly for $Q$ matrix. After the matrix calculation, Eq. 11 fuses the information from two embeddings while keeping some unique information from the special $[class]$ token.

$$H = softmax(\frac{QK^T}{\sqrt{d_k}})V \tag{9}$$

$$K_1 = [K_{class_1}, K_1 + K_2], Q_1 = [Q_{class_1}, Q_1 + Q_2] \tag{10}$$

$$Q_1 K_1^T = \begin{bmatrix} Q_{class_1} K_{class_1} & (Q_1 + Q_2) K_{class_1} \\ Q_{class_1}(K_1 + K_2) & (Q_1 + Q_2)(K_1 + K_2) \end{bmatrix} \tag{11}$$

## 3.3 Morphology Augmentation

A key characteristic of the AD-plagued brain is that, as the disease progresses, an increasing amount of brain mass will suffer from atropy. When this process is reflected in brain imaging, the there will

Figure 2: The visualization of Alzheimer's Disease (AD) image, Normal Control (NC) image and Mild Cognitive Impairment (MCI) image. The left is the raw image and the right is the augmented image. Such that for the two images in the third blue border (MCI to AD), an MCI image (left) is augmented by **Morphology Augmentation** into AD (right) and classified as AD for model training. The cerebral ventricle (red circle) has a significant difference in size for AD and NC.

be empty "holes" of the brain if one has AD. Based on this, we propose a morphology augmentation, an augmentation method which help to expand and reduce the size of the atrophy, causing the improvement of the model. This augmentation is based on atrophy expansion and atrophy reduction shown in Eq. 12, 13. $f$ is the input image, $b_N$ is the atrophy expansion or atrophy reduction element, $(x,y)$ and $(s,t)$ are the coordinates in $f$ and $b_N$ respectively.

$$[f \ominus b_N](x,y) = \min_{(s,t) \in b_N} \{f(x+s, y+t) - b_N(s,t)\} \tag{12}$$

$$[f \oplus b_N](x,y) = \max_{(s,t) \in b_N} \{f(x-s, y-t) + b_N(s,t)\} \tag{13}$$

We apply atropy expansion augmentation to AD images and MCI images and label the resultant images as AD; on the other hand, we apply atropy reduction augmentation to Normal Control(NC) images and MCI images and label the resultant images as NC, where MCI is the prodromal stage of AD. The visualization of morphology augmentation is shown in Fig. 2.

### 3.4 ADAPTIVE TRAINING STRATEGY

To further investigate the potential of ADAPT, we propose an attention score based training strategy in order to allow our model to extract more features from the more important dimension with limited size of inputs. We calculate the attention score of each dimension after the final inter-dimension cross attention encoder layer according to Eq. 14. Because our $[class]$ token is dimension specific, so we just calculate the attention score of the $[class]$ token as the representation of the special dimension. This strategy allows our network to adaptively choose the slice number of each dimension while updating itself.

$$H_{dim} = softmax(\frac{Q_{class_{dim}} K^T}{\sqrt{d_k}})V \tag{14}$$

---

**Algorithm 1** ADAPT Training Strategy

---

**Input:** 3D MRI Training set $T$, initial slice number list $\psi$, model ADAPT $\Theta$, total slice number $n_{total}$
**Output:** Updated model $\Theta$, final list $\psi$
1: **while** $Training$ **do**
2:     With $T$ and $\psi$, sample 2D data $\delta_a, \delta_c \delta_s$ on axial, coronal and sagittal views.
3:     $\delta_a, \delta_c, \delta_s$ = SAE($\delta_a, \delta_c, \delta_s$)
4:     $\delta_a, \delta_c, \delta_s$ = IntraCAE$_a$(DSAE$_a$($\delta_a$)), IntraCAE$_c$(DSAE$_c$($\delta_c$)), IntraCAE$_s$(DSAE$_s$($\delta_s$))
5:     $\delta_a, \delta_c, \delta_s$ = InterCAE($\delta_a, \delta_c, \delta_s$)
6:     Calculate score using Eq. 14 for three dimensions
7:     Calculate cross-entropy loss and update $\Theta$
8:     **if** p **then**
9:         Update $\psi$ according to Eq 15.
10:     **else**
11:         INITIALIZE($\psi$)
12:     **end if**
13: **end while**

---

We then adaptively update the slice number of each dimension based on normalized attention scores using Eq. 15, where $n$ is the total slice number and $\psi$ is the slice number list. Here we also constrain

| Model name | Model size (#params) | GFLOPs | Morphology Aug | ADNI val acc. | ADNI test acc. | AIBL test acc. | MIRIAD test acc. | OASIS test acc. |
|---|---|---|---|---|---|---|---|---|
| MedicalNet-10 | 17,723,458 | 225.7 | No | 0.855 ± 0.015 | 0.851 ± 0.016 | 0.880 ± 0.007 | 0.845 ± 0.007 | 0.802 ± 0.002 |
| **MedicalNet-10 Chen et al. (2019)** | **17,723,458** | **225.7** | **Yes** | 0.827 ± 0.013 | 0.811 ± 0.009 | 0.808 ± 0.011 | **0.849**±0.016 | 0.752 ± 0.013 |
| MedicalNet-18 | 36,527,938 | 492.6 | No | 0.772 ± 0.005 | 0.750 ± 0.006 | 0.874 ± 0.012 | 0.815 ± 0.010 | 0.801 ± 0.003 |
| **MedicalNet-18 Chen et al. (2019)** | **36,527,938** | **492.6** | **Yes** | 0.739 ± 0.008 | **0.782**±0.002 | 0.757 ± 0.009 | **0.896**±0.012 | 0.742 ± 0.010 |
| MedicalNet-34 | 66,837,570 | 910.8 | No | 0.622 ± 0.005 | 0.635 ± 0.006 | 0.660 ± 0.012 | 0.704 ± 0.010 | 0.546 ± 0.003 |
| **MedicalNet-34 Chen et al. (2019)** | **66,837,570** | **910.8** | **Yes** | **0.635**±0.018 | **0.691**±0.012 | **0.727**±0.010 | **0.805**±0.006 | **0.711**±0.004 |
| MedicalNet-50 | 59,626,818 | 666.8 | No | 0.639 ± 0.012 | 0.650 ± 0.006 | 0.705 ± 0.007 | 0.742 ± 0.011 | 0.649 ± 0.014 |
| **MedicalNet-50 Chen et al. (2019)** | **59,626,818** | **666.8** | **Yes** | 0.612 ± 0.013 | 0.525 ± 0.014 | 0.614 ± 0.007 | 0.673 ± 0.004 | **0.660**±0.013 |
| MedicalNet-101 | 98,672,962 | 1181.1 | No | 0.619 ± 0.012 | 0.587 ± 0.015 | 0.674 ± 0.007 | 0.647 ± 0.003 | 0.585 ± 0.005 |
| **MedicalNet-101 Chen et al. (2019)** | **98,672,962** | **1181.1** | **Yes** | 0.571 ± 0.005 | **0.626**±0.004 | **0.729**±0.017 | **0.675**±0.007 | **0.628**±0.005 |
| MedicalNet-152 | 130,831,682 | 1604.8 | No | 0.536 ± 0.014 | 0.540 ± 0.006 | 0.604 ± 0.009 | 0.560 ± 0.006 | 0.490 ± 0.009 |
| **MedicalNet-152 Chen et al. (2019)** | **130,831,682** | **1604.8** | **Yes** | **0.543**±0.015 | **0.632**±0.007 | **0.730**±0.007 | **0.655**±0.017 | **0.626**±0.002 |
| 3D Resnet-34 | 63,470,658 | 341.1 | No | 0.540 ± 0.007 | 0.572 ± 0.009 | 0.545 ± 0.012 | 0.584 ± 0.005 | 0.492 ± 0.004 |
| **3D Resnet-34 He et al. (2016)** | **63,470,658** | **341.1** | **Yes** | **0.560**±0.005 | **0.587**±0.008 | **0.652**±0.017 | **0.661**±0.010 | **0.504**±0.008 |
| 3D Resnet-50 | 46,159,170 | 256.9 | No | 0.540 ± 0.007 | 0.572 ± 0.009 | 0.545 ± 0.012 | 0.584 ± 0.005 | 0.492 ± 0.004 |
| **3D Resnet-50 He et al. (2016)** | **46,159,170** | **256.9** | **Yes** | **0.560**±0.005 | **0.587**±0.008 | **0.652**±0.017 | **0.652**±0.017 | **0.504**±0.008 |
| 3D Resnet-101 | 85,205,314 | 391.1 | No | 0.556 ± 0.011 | 0.468 ± 0.014 | 0.601 ± 0.008 | 0.590 ± 0.015 | 0.537 ± 0.014 |
| **3D Resnet-101 He et al. (2016)** | **85,205,314** | **391.1** | **Yes** | **0.560**±0.009 | **0.587**±0.008 | **0.652**±0.010 | **0.661**±0.012 | **0.504**±0.009 |
| 3D DenseNet-121 | 11,244,674 | 260.5 | No | 0.591 ± 0.001 | 0.545 ± 0.004 | 0.651 ± 0.012 | 0.670 ± 0.005 | 0.699 ± 0.007 |
| **3D DenseNet-121 Huang et al. (2017)** | **11,244,674** | **260.5** | **Yes** | 0.576 ± 0.009 | **0.620**±0.005 | **0.781**±0.005 | 0.375 ± 0.011 | **0.744**±0.004 |
| 3D DenseNet-201 | 25,334,658 | 286.5 | No | 0.584 ± 0.005 | 0.605 ± 0.007 | 0.644 ± 0.008 | 0.540 ± 0.014 | 0.653 ± 0.007 |
| **3D DenseNet-201 Huang et al. (2017)** | **25,334,658** | **286.5** | **Yes** | 0.552 ± 0.003 | **0.620**±0.007 | **0.691**±0.015 | **0.385**±0.006 | **0.674**±0.014 |
| Knowledge4D | 33,162,880 | 633.9 | No | 0.605 ± 0.005 | 0.716 ± 0.003 | 0.764 ± 0.002 | 0.650 ± 0.002 | 0.799 ± 0.006 |
| **Knowledge4D Zhou et al. (2023b)** | **33,162,880** | **633.9** | **Yes** | 0.515 ± 0.010 | 0.617 ± 0.011 | **0.789**±0.002 | 0.435 ± 0.005 | 0.744 ± 0.004 |
| I3D | 12,247,332 | 191 | No | 0.466 ± 0.008 | 0.612 ± 0.005 | 0.630 ± 0.008 | 0.537 ± 0.012 | 0.597 ± 0.007 |
| **I3D Carreira & Zisserman (2017)** | **12,247,332** | **191** | **Yes** | 0.465 ± 0.010 | **0.643**±0.007 | **0.680**±0.005 | **0.549**±0.007 | **0.613**±0.012 |
| FCNlinksCNN | 310,488,372 | 375.6 | No | 0.572 ± 0.008 | 0.453 ± 0.005 | 0.303 ± 0.008 | 0.718 ± 0.012 | 0.562 ± 0.007 |
| **FCNlinksCNN Qiu et al. (2020)** | **310,488,372** | **375.6** | **Yes** | 0.536 ± 0.008 | **0.474**±0.016 | **0.477**±0.004 | **0.743**±0.006 | **0.563**±0.011 |
| COVID-ViT | 78,177,282 | 448.6 | No | 0.515 ± 0.004 | 0.553 ± 0.007 | 0.543 ± 0.012 | 0.338 ± 0.002 | 0.682 ± 0.008 |
| **COVID-ViT Gao et al. (2021)** | **78,177,282** | **448.6** | **Yes** | 0.500 ± 0.012 | **0.569**±0.013 | **0.630**±0.014 | **0.38**±0.002 | **0.720**±0.011 |
| Uni4Eye | 340,324,866 | 78.4 | No | 0.519 ± 0.002 | 0.597 ± 0.017 | 0.655 ± 0.011 | 0.343 ± 0.004 | 0.713 ± 0.009 |
| **Uni4Eye Cai et al. (2022)** | **340,324,866** | **78.4** | **Yes** | **0.521**±0.007 | **0.620**±0.011 | **0.740**±0.012 | 0.340±0.005 | **0.755**±0.012 |
| ADAPT | 9,695,490 | 46.3 | No | 0.842 ± 0.005 | 0.862 ± 0.007 | 0.905 ± 0.003 | 0.853 ± 0.007 | 0.818 ± 0.009 |
| **ADAPT** | **9,695,490** | **46.3** | **Yes** | **0.900**±0.009 | **0.920**±0.002 | **0.921**±0.004 | **0.907**±0.005 | **0.864**±0.002 |

Table 1: Comparison of accuracy various 3D CNN-based and transformer-based models on multi-institutional Alzheimer's disease dataset. The numerical numbers of models with morphology augmentation are bolded when getting better performance.

the selection pool to make sure the model will attend across multiple attentions. The full training strategy is shown in algorithm 1. To avoid the model will stick with certain view dimension after the first choice, we also allow the model to change the attention with certain probabilities p.

$$n_{dim} = round(\frac{\hat{H_{dim}}}{\sum_{r \in \psi} r} * n_{total}), \quad n_{dim} = \begin{cases} n_{min}, n_{dim} \geq n_{min} \\ n_{max}, n_{dim} \leq n_{max} \end{cases} \quad (15)$$

# 4 EXPERIMENTS

## 4.1 EXPERIMENTAL SETTINGS

**Implementation Details.** We implemented ADAPT using a Pytorch library Paszke et al. (2019). ADAPT was trained using an AdamW optimizer with a learning rate of 0.00005. All other parameters are default. At the same time, we also took the advantage of cosine learning rate from Loshchilov & Hutter (2016). We treat this as a binary classification task, so we use cross-entropy loss Zhang & Sabuncu (2018). The training process used 2 80G NVIDIA A800 GPUs. Due to the memory capacity, we use 6 batches on each GPU, meaning a total batch size of 12. We also do data preprocessing, the details are in Appendix A.1.

**Datasets.** To verify the effectiveness of our ADAPT, we use the dataset from the Alzheimer's Disease Neuroimaging Initiative (ADNI) for the training process. Then we also evaluate our trained ADAPT and other baselines with AIBL, MIRIAD and OASIS datasets. The details of these datasets can be found in Appendix A.2. Each of the images for any dataset is a 3D grayscale image.

## 4.2 EVALUATION BETWEEN BASELINES

Our ADAPT was compared with various baseline models, including 3D CNN-based models: 3D DenseNet (121, 201) Huang et al. (2017), 3D ResNet (34, 50, 101) He et al. (2016) because they have been widely used for AD classification Korolev et al. (2017); Ruiz et al. (2020); Yang et al. (2018); Zhang et al. (2021). We also add other baselines to show the capability of our ADAPT, including: MedicalNet Chen et al. (2019), I3D Carreira & Zisserman (2017), FCNlinksCNN Qiu et al. (2020) and Knowledge4D Zhou et al. (2023b). Each MedicalNet is based on a basic Resnet He et al. (2016)

model, such that MedicalNet-10 is based on Resnet-10 respectively. We also compare our method with 3D transformer-based models: COVID-VIT Gao et al. (2021), Uni4Eye Cai et al. (2022).

In the experiment, we chose 48 slices as input, meaning 16 equidistant slices on each view as initial. Because we found the central part of 3D images would be more important and consist of more useful information, we applied the **important sampling** method in our slice-picking stage. To be more specific, for a $224\times224\times224$ image, we pick equidistant slices from $52^{nd}$ to $172^{nd}$ on each view.

The experiment is conducted for three times and the quantitative performance is presented in Table 1. We choose the model with the best validation accuracy on ADNI and then test it on various Alzheimer's disease datasets. This kind of method can verify if the model has learned well on the knowledge that is highly transferable across different datasets. We also record the total parameters and GFLOPs of each model. We set up an ablation study on Morphology Augmentation to test the effectiveness. Overall, ADAPT achieves the best performance on i.i.d testing scenario (ADNI) as well as all out-of-domain testing scenarios (AIBL, MIRIAD and OASIS). We believe these results show that ADAPT is not only superior in Alzheimer's diagnosis in i.i.d setting, but also fairly robust when the testing data is collected from different facilities. At the same time, our model has the least parameters and GLOPs, demonstrating the success of our novel method in attacking the AD diagnosing task using 2D based model.

The best performance is achieved when ADAPT chooses 14 slices from saggital view, and 17 slices from the coronal and axial view respectively. As compared with table 4, we found the interesting facts that coronal and axial view may contain more differential relationships about cortex and ventricle of AD and NC, which can help the model learn the special attention features accurately.

By analyzing the morphology augmentation result (bolded one), we found that it can greatly improve the diagnosis accuracy on most models. However, for the Medicalnet with fewer layers, the augmentation method cannot guarantee improvement. These are due to the following two reasons:

- The morphology augmentation method enlarges the dataset with the MCI data included. The small CNN-based models will be overfitting quickly when trained with large dataset. However, the transformer-based models usually need more data to be trained sufficiently, thus morphology augmentation will show its power when applying transformer-based models to alzheimer's disease diagnosis.

- CNN-based models rely on local bias detection to do diagnosis. Morphology augmentation may melt some of the cortex details but augment the atrophy (see Fig 2). This may cause the lost of some local details.

### 4.3 ABLATION STUDY

To evaluate how effective each block is, we compared our ADAPT with other variants, changing one setting each time. We first changed the transformer attention layers of each encoder. We investigate how the number of layers will affect our ADAPT performance. The results are shown in Table 2, there are four numbers in each variant, each one corresponding to an encoder block. Such as 1+1+2+2 meaning that the shared self-attention encoders, dimension-specific self-attention encoders, intra-dimension cross-attention encoders and inter-dimension cross-attention encoders have 1, 1, 2, 2 transformer attention layer respectively. The result shows that ADAPT outperforms all the variants on test accuracy in all four datasets.

| Layer Number | ADNI | | AIBL | MIRIAD | OASIS |
|---|---|---|---|---|---|
| | Val acc. | Test acc. | Test acc. | Test acc. | Test acc. |
| 1+1+1+1 | 0.713 | 0.776 | 0.800 | 0.685 | 0.793 |
| 2+2+1+1 | 0.770 | 0.811 | 0.863 | 0.903 | 0.716 |
| 2+2+2+2 | 0.881 | 0.911 | 0.897 | 0.669 | 0.800 |
| 3+3+3+3 | 0.917 | 0.895 | 0.907 | 0.723 | 0.806 |
| **Ours (1+1+2+2)** | 0.9 | **0.920** | **0.921** | **0.907** | **0.864** |

Table 2: Comparison of accuracy between ADAPT and four variants ablating with different numbers of transformer layers in each encoder in the four datasets.

| Cross-Attention Mechanism | ADNI | | AIBL | MIRIAD | OASIS |
|---|---|---|---|---|---|
| | Val acc. | Test acc. | Test acc. | Test acc. | Test acc. |
| No Cross-Attention | 0.719 | 0.627 | 0.810 | 0.710 | 0.675 |
| Class Token Cross-Attention | 0.878 | 0.848 | 0.864 | 0.709 | 0.606 |
| Easy Concat Cross-Attention | 0.917 | 0.783 | 0.723 | 0.806 | 0.681 |
| **Ours (Fusion Attention)** | 0.9 | **0.920** | **0.921** | **0.907** | **0.864** |

Table 3: Comparison of accuracy between ADAPT and three variants ablating different cross attention mechanisms in the four datasets.

| Models | ADNI | | AIBL | MIRIAD | OASIS |
|---|---|---|---|---|---|
| | Val acc. | Test acc. | Test acc. | Test acc. | Test acc. |
| w/o Adaptive Training | 0.855 | 0.836 | 0.883 | 0.882 | 0.807 |
| w/o Guide Embedding | 0.880 | 0.860 | 0.863 | 0.869 | 0.826 |
| w/o Torchio | 0.878 | 0.876 | 0.899 | 0.864 | 0.802 |
| w/o Important Sampling | 0.823 | 0.886 | 0.852 | 0.887 | 0.838 |
| **ADAPT** | 0.9 | **0.920** | **0.921** | **0.907** | **0.864** |

Table 4: Comparison of accuracy between ADAPT and four variants ablating different training augmentation settings in the four datasets.

| Component | ADNI | | AIBL | MIRIAD | OASIS |
|---|---|---|---|---|---|
| | Val acc. | Test acc. | Test acc. | Test acc. | Test acc. |
| w/o SAE | 0.872 | 0.905 | 0.914 | 0.869 | 0.848 |
| w/o DS-AE | 0.859 | 0.859 | 0.901 | 0.781 | 0.817 |
| w/o IntraCAE | 0.885 | 0.867 | 0.904 | 0.885 | 0.827 |
| w/o InterCAE | 0.872 | 0.864 | 0.877 | 0.87 | 0.851 |
| **ADAPT** | 0.9 | **0.920** | **0.921** | **0.907** | **0.864** |

Table 5: Comparison of accuracy between ADAPT and four variants ablating different main components of ADAPT architecture in the four datasets.

Table 3 shows how different cross-attention mechanisms will affect the final result. The first variant: No Cross-Attention, meaning that we didn't apply any cross-attention mechanism in the last two encoder blocks. Class Token Cross-Attention is a variant of Eq. 10. It adds the $[class]$ token embedding up but not the embedding behind the $[class]$ token. For the easy concat cross-attention mechanism, it simply concatenates the embeddings from different slices and view dimensions into a whole large embedding. Our proposed Fusion Attention achieves more than 7% improvements to the ADNI test result while demonstrating superiority on other testing datasets, verifying that fusion attention cannot only fuse the information while keeping the unique information in each embedding.

Table 4 shows other variables in our settings. We delete one important setting in each variant to see the results. ADAPT outperforms all variant models in all four datasets by 3.2%, 7.1%, 3.8% and 6.0%, respectively. The results show the great capability of different settings in augmenting the model learning ability to classify 3D MRI.

Table 5 shows the results after ablating the main components of ADAPT one by one. We can find that each component is indispensable and vital for the final performance of ADAPT. In conclusion, DS-AE block will contribute the most, because it plays the role of extracting detailed features from each 2D slice from different views, not only leading the whole model to focus on special features but also guiding the adaptive training strategy to determine which view is more important.

### 4.4 VISUALIZATION RESULT

We visualize the activated area of our model based on the transformer attention map. Figure 3 shows a NC-related attention map in 3D MRI images from ADNI dataset in sagittal, coronal and axial views. Because ADAPT has 4 special encoders, we visualize the attention result after each encoder.

We found that for NC and AD result, the attention mostly focused on some special brain tissues, such as hippocampus, cortex, ventricle and frontal lobe. Disruption of the frontal lobes and its associated networks are a common consequence of neurodegenerative disorders Sawyer et al. (2017), as well as the hippocampus is most notably damaged by AD Xu et al. (2021). Based on these understandings of Alzheimer's pathology Frisoni et al. (2010), ADAPT successfully captured the AD-related part

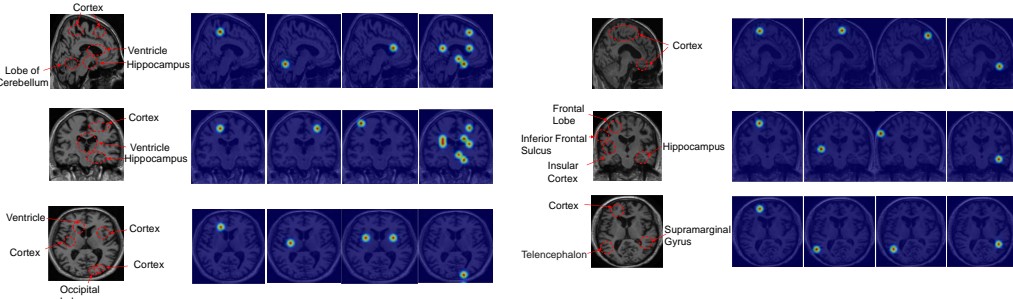

Figure 3: Attention map for Normal Control result. Each line corresponds to one view dimension: saggital, coronal and axial.

Figure 4: Attention map for Alzheimer's Disease result. Each line corresponds to one view dimension: saggital, coronal and axial.

because with the procedure of Alzheimer's, the hippocampus and cortex begin to atrophy, and the ventricle begins to expand, which can serve as an evidence of morphology augmentation and confirm the reliability of our proposed ADAPT.

## 5 CONCLUSIONS

We proposed a 3D medical image classification model, called ADAPT, that uses various 2D transformer encoder blocks for Alzheimer's disease diagnosis. The proposed method uses shared self-attention encoders across different view dimensions, dimension-specific self-attention encoders, intra-dimension cross-attention encoders, and inter-dimension cross-attention encoders to extract and combine information from high-dimensional 3D MRI images, with novel techniques such as fusion attention mechanism and morphology augmentation. With different encoders, our adaptive training strategy can allow physicians to pay more attention to different dimensions of MRI images. The experiments show that ADAPT can achieve outstanding performance while utilizing the least memory compared to various 3D image classification networks in multi-institutional test datasets. The visualization results show that ADAPT can successfully focus on AD-related regions of 3D MRI images, guiding accurate and efficient clinical research on Alzheimer's Disease.

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

## A  ALZHEIMER'S DIAGNOSIS EXPERIMENTS

### A.1  IMPLEMENTATION DETAILS

We implement consistent data pre-processing techniques to normalize and standardize MRI images sourced from a multi-institutional database. We first do data augmentation in the following steps. we have followed closely the recommended protocol from the medical community Wen et al. (2020) to process the data. Firstly, we do bias field correction with N4ITK method Tustison et al. (2010). Next, we register each image to the MNI space Fonov et al. (2009; 2011) with the ICBM 2009c nonlinear symmetric template by performing a affine registration using the SyN algorithm Avants et al. (2014) from ANTs Avants et al. (2008). At the same time, the registered images were further cropped to remove the background to improve the computational efficiency. These operations result in 1 mm isotropic voxels for each image. Intensity rescaling, which was performed based on the minimum and maximum values, denoted as MinMax, was also set to be optional to study its influence on the classification results. Finally, the deep QC system Fonov et al. (2018) is performed to check the quality of the linearly registered data. The software outputs a probability indicating how accurate the registration is. We excluded the scans with a probability lower than 0.5. Overall, the registration process we perform on the data maps different sets of images into a single coordinate system to prepare the data for our later usage.

We also use the Torchio library Pérez-García et al. (2021) in the training set. Meanwhile, we resize all the MRI images with Scipy library Virtanen et al. (2020) into $224 \times 224 \times 224$ to better fit the input of our ADAPT. Finally, we employed the zero-mean unit-variance method to normalize the intensity of all voxels within the images.

For the training dataset, we apply morphology augmentation to the same MCI data, classify the MCI into NC after doing atrophy reduction augmentation, and classify it into AD after doing atrophy expansion augmentation. In this way, each MCI is used twice, significantly enlarging the dataset. At the same time, we also do morphology augmentation to AD and NC images randomly, with a probability of 0.5.

After preprocessing the 3D MRI images, we cut them into 2D slices along sagittal, coronal and axial views. Then we choose 16 slices in each view as the initial data and concatenate them into a sequence. We choose equidistant slices on each view and embed them into patch embedding similar to ViT. Here we choose the embed layer from Touvron et al. (2022). Then we use a total of 6 standard transformer attention layers, and 1 layer for each of the first two encoders, 2 layers for each of the last two encoders, with 4 heads. For the adaptive training strategy, we set the probability p as 0.8. At last, because we have three $[class]$ tokens, each representing a special view dimension, we use a classification MLP head, with input feature number $3 \times 256$ and output feature number 2, aiming to figure out whether the image is from a disease or not.

### A.2  DATASETS DESCRIPTION

The ADNI dataset consists of MRI images of T1-weighted magnetic resonance imaging subjects. There are a total of 3,891 3D MRI images in the dataset, including 1,216 normal cases (NC), 1,110 AD cases and 1,565 MCI cases. During the training, 878 normal images, 884 AD images and 1565 MCI images were split into the training set, with 72 normal images and 81 AD images as a validation set, together with 266 normal images and 145 AD images as a testing set. All splits have no overlapping subjects.

Meanwhile, to evaluate the performance of our ADAPT and other deep learning baseline models, we also consider other datasets as test sets. We mainly acquire them from three other institutions with the ADNI test dataset: Australian Imaging, Biomarker and Lifestyle Flagship Study of Ageing (AIBL), Minimal Interval Resonance Imaging in Alzheimer's Disease (MIRIAD), and The Open Access Series of Imaging Studies (OASIS). The AIBL dataset contains a total of 413 images with 363 NC and 50 AD after dropping all MCI cases. The MIRIAD dataset contains a total of 523 cases which consist of 177 NC and 346 AD cases. The OASIS dataset contains a total of 2157 cases which consist of 1692 NC and 465 AD cases.

| Model name | | ADNI | | | | | | AIBL | | | MIRIAD | | | OASIS | | |
|---|---|---|---|---|---|---|---|---|---|---|---|---|---|---|---|---|
| | | Valid | | | Test | | | Test | | | Test | | | Test | | |
| | | brier | specificity | roc | brier | specificity | roc | brier | specificity | roc | brier | specificity | roc | brier | specificity | roc |
| MedicalNet-10 | w/o Aug | 0.314 | 0.852 | 0.852 | 0.452 | 0.795 | 0.799 | 0.710 | 0.449 | 0.663 | 0.266 | 0.852 | 0.848 | 0.575 | 0.456 | 0.630 |
| **MedicalNet-10** | **Aug** | 0.413 | 0.827 | 0.824 | 0.628 | 0.790 | **0.801** | 0.787 | **0.673** | **0.736** | 0.360 | **0.898** | **0.873** | 0.603 | **0.669** | **0.709** |
| MedicalNet-18 | w/o Aug | 0.421 | 0.790 | 0.783 | 0.325 | 0.835 | 0.774 | 0.778 | 0.359 | 0.616 | 0.342 | 0.869 | 0.841 | 0.629 | 0.363 | 0.583 |
| **MedicalNet-18** | **Aug** | **0.266** | 0.713 | 0.726 | 0.466 | 0.786 | **0.786** | **0.617** | **0.696** | **0.726** | 0.195 | **0.887** | **0.873** | 0.609 | **0.689** | **0.716** |
| MedicalNet-34 | w/o Aug | 0.194 | 0.585 | 0.603 | 0.217 | 0.666 | 0.651 | 0.177 | 0.547 | 0.603 | 0.167 | 0.452 | 0.578 | 0.193 | 0.619 | 0.584 |
| **MedicalNet-34** | **Aug** | **0.188** | 0.582 | **0.609** | 0.336 | **0.711** | **0.701** | 0.467 | **0.654** | **0.688** | 0.147 | **0.730** | **0.761** | **0.187** | **0.626** | **0.673** |
| MedicalNet-50 | w/o Aug | 0.247 | 0.605 | 0.623 | 0.399 | 0.764 | 0.706 | 0.662 | 0.683 | 0.694 | 0.134 | 0.571 | 0.657 | 0.528 | 0.703 | 0.680 |
| **MedicalNet-50** | **Aug** | **0.230** | 0.508 | 0.561 | **0.269** | 0.648 | 0.586 | **0.119** | **0.757** | 0.591 | 0.142 | 0.349 | 0.510 | **0.041** | **0.775** | 0.637 |
| MedicalNet-101 | w/o Aug | 0.18 | 0.567 | 0.591 | 0.425 | 0.488 | 0.54 | 0.303 | 0.421 | 0.513 | 0.139 | 0.432 | 0.54 | 0.365 | 0.508 | 0.546 |
| **MedicalNet-101** | **Aug** | **0.156** | 0.507 | 0.539 | 0.576 | **0.62** | **0.543** | 0.458 | 0.219 | 0.511 | 0.155 | 0.374 | 0.524 | 0.764 | 0.44 | 0.535 |
| MedicalNet-152 | w/o Aug | 0.018 | 0.467 | 0.503 | 0.519 | 0.445 | 0.493 | 0.242 | 0.411 | 0.508 | 0.245 | 0.598 | 0.579 | 0.243 | 0.561 | 0.509 |
| **MedicalNet-152** | **Aug** | 0.04 | **0.469** | **0.507** | 0.45 | **0.644** | **0.523** | 0.65 | **0.43** | 0.488 | 0.062 | **0.657** | 0.566 | 0.52 | 0.392 | **0.511** |
| 3D ResNet-34 | w/o Aug | 0.238 | 0.478 | 0.511 | 0.258 | 0.484 | 0.529 | 0.258 | 0.448 | 0.498 | 0.245 | 0.571 | 0.576 | 0.258 | 0.568 | 0.531 |
| **3D ResNet-34** | **Aug** | **0.188** | 0.494 | 0.526 | 0.237 | 0.58 | 0.543 | 0.329 | 0.744 | 0.525 | 0.118 | 0.584 | 0.5 | 0.526 | 0.617 | 0.521 |
| 3D ResNet-50 | w/o Aug | 0.237 | 0.521 | 0.546 | 0.256 | 0.458 | 0.446 | 0.253 | 0.491 | 0.511 | 0.257 | 0.574 | 0.554 | 0.317 | 0.353 | 0.494 |
| **3D ResNet-50** | **Aug** | 0.218 | 0.499 | 0.519 | 0.353 | 0.413 | **0.495** | 0.335 | 0.247 | **0.521** | 0.262 | 0.530 | 0.554 | 0.317 | 0.353 | 0.494 |
| 3D ResNet-101 | w/o Aug | 0.252 | 0.518 | 0.538 | 0.302 | 0.458 | 0.463 | 0.345 | 0.651 | 0.456 | 0.222 | 0.536 | 0.516 | 0.337 | 0.611 | 0.497 |
| **3D ResNet-101** | **Aug** | 0.207 | 0.476 | 0.514 | 0.306 | 0.449 | **0.476** | 0.396 | 0.244 | **0.460** | 0.217 | 0.427 | 0.550 | 0.342 | **0.750** | 0.510 |
| 3D DenseNet-121 | w/o Aug | 0.243 | 0.535 | 0.563 | 0.242 | 0.628 | 0.565 | 0.24 | 0.58 | 0.616 | 0.274 | 0.82 | 0.747 | 0.233 | 0.632 | 0.662 |
| **3D DenseNet-121** | **Aug** | **0.243** | 0.483 | 0.529 | 0.262 | 0.615 | 0.512 | 0.241 | 0.179 | 0.481 | 0.346 | 0.335 | 0.5 | 0.261 | 0.241 | 0.492 |
| 3D DenseNet-201 | w/o Aug | 0.228 | 0.522 | 0.553 | 0.268 | 0.579 | 0.517 | 0.282 | 0.594 | 0.512 | 0.263 | 0.76 | 0.576 | 0.273 | 0.824 | 0.521 |
| **3D DenseNet-201** | Aug | **0.169** | 0.494 | 0.523 | 0.414 | 0.435 | **0.523** | 0.379 | **0.676** | **0.557** | 0.286 | 0.614 | **0.597** | 0.389 | 0.753 | **0.524** |
| Knowledge4D | w/o Aug | 0.296 | 0.622 | 0.612 | 0.345 | 0.628 | 0.672 | 0.399 | 0.45 | 0.608 | 0.332 | 0.817 | 0.732 | 0.392 | 0.472 | 0.632 |
| **Knowledge4D** | **Aug** | **0.249** | 0.507 | 0.51 | 0.43 | 0.548 | 0.565 | 0.462 | 0.411 | **0.633** | 0.379 | 0.698 | 0.567 | 0.442 | **0.637** | 0.629 |
| I3D | w/o Aug | 0.257 | 0.508 | 0.488 | 0.267 | 0.461 | 0.537 | 0.274 | 0.544 | 0.587 | 0.237 | 0.538 | 0.536 | 0.273 | 0.488 | 0.542 |
| **I3D** | **Aug** | 0.335 | **0.538** | 0.5 | 0.339 | 0.354 | 0.518 | 0.275 | 0.524 | **0.602** | 0.237 | 0.56 | 0.554 | 0.273 | 0.487 | 0.548 |
| FCNlinksCNN | w/o Aug | 0.233 | 0.527 | 0.549 | 0.23 | 0.668 | 0.562 | 0.221 | 0.783 | 0.542 | 0.217 | 0.489 | 0.604 | 0.222 | 0.72 | 0.64 |
| **FCNlinksCNN** | **Aug** | **0.229** | 0.481 | 0.51 | 0.233 | 0.666 | **0.571** | 0.2298 | 0.635 | **0.556** | 0.232 | **0.629** | 0.687 | 0.225 | **0.774** | 0.599 |
| COVID-ViT | w/o Aug | 0.252 | 0.519 | 0.516 | 0.245 | 0.503 | 0.529 | 0.238 | 0.588 | 0.568 | 0.257 | 0.662 | 0.5 | 0.24 | 0.345 | 0.513 |
| **COVID-ViT** | **Aug** | 0.251 | 0.528 | 0.518 | 0.251 | 0.535 | 0.592 | 0.248 | **0.635** | **0.633** | 0.221 | 0.662 | 0.5 | 0.256 | 0.384 | 0.562 |
| Uni4Eye | w/o Aug | 0.264 | 0.564 | 0.542 | 0.274 | 0.431 | 0.513 | 0.277 | 0.492 | 0.473 | 0.241 | 0.55 | 0.496 | 0.243 | 0.561 | 0.509 |
| **Uni4Eye** | **Aug** | 0.346 | 0.598 | 0.554 | 0.325 | 0.436 | 0.527 | 0.586 | 0.56 | 0.528 | 0.559 | 0.562 | 0.5 | 0.482 | 0.582 | 0.547 |
| ADAPT | w/o Aug | 0.377 | 0.818 | 0.828 | 0.672 | 0.748 | 0.805 | 0.692 | 0.641 | 0.776 | 0.434 | 0.886 | 0.831 | 0.47 | 0.58 | 0.724 |
| **ADAPT** | **Aug** | 0.371 | **0.918** | **0.909** | 0.659 | **0.855** | **0.887** | 0.684 | **0.650** | **0.787** | 0.210 | 0.850 | **0.876** | 0.580 | **0.603** | **0.732** |

Table 6: Comparison of brier score, specificity score and ROC-AUC score various 3D CNN-based and transformer-based models on multi-institutional Alzheimer's disease dataset. The numerical numbers of models with morphology augmentation are bolded when getting better performance.

### A.3 MULTI-METRICS PERFORMANCE

Considering that the Alzheimer's experimental datasets are usually imbalanced, we also verify the performance of ADAPT using other metrics, including brier score Rufibach (2010), specificity score Glaros & Kline (1988), and ROC-AUC score Hoo et al. (2017), which are usually used in clinical research. Table 6 shows the detailed results of ADAPT and various baselines on multi-institutional Alzheimer's disease dataset. We observe that the conclusion is consistent with Section 4.2. ADAPT can still achieve the best ROC-AUC score compared to all the baselines. Also the specificity score is also the best on ADNI dataset, meaning that ADAPT can accurately classify the negative samples. The morphology augmentation greatly improves the performance of transformer-based models. At the same time, after applying the augmentation method, the ROC-AUC score was improved on most of the models, including CNN-based ones. These metrics also reflect the power of our proposed ADAPT and morphology augmentation.

## B GLIOBLASTOMA SUBTYPE DIAGNOSIS

A malignant brain tumor, known as glioblastoma, is a life-threatening condition. It is the most common and deadliest form of brain cancer in adults, with a median survival time of less than a year. The presence of MGMT promoter methylation, a specific genetic sequence in the tumor, has been identified as a favorable prognostic factor and a strong predictor of responsiveness to chemotherapy. We tried to use ADAPT to predict the genetic subtype of glioblastoma, which will potentially minimize the number of surgeries and refine the type of therapy required.

### B.1 DATASET DESCRIPTION

We collected the brain tumor dataset Baid et al. (2021), which consists of 585 MRI samples and classified into two subtypes. We resized the T1-weighted post-contrast multi-parametric MRI (mpMRI) scans into 224 pixels and use the resized gray-scale image to construct the 3D volume data. Then we split it into train, validation and test sets according to the ratio of 8:1:1. In conclusion, there are 226 subtype *0* and 242 subtype *1* in training set, 27 subtype *0* and 31 subtype *1* in validation

| Model name | | Tumor | | | | | | | |
|---|---|---|---|---|---|---|---|---|---|
| | | Valid | | | | Test | | | |
| | | acc. | brier | specificity | roc | acc. | brier | specificity | roc |
| MedicalNet-10 | w/o Aug | 0.621 | 0.592 | 0.617 | 0.619 | 0.433 | 0.488 | 0.412 | 0.423 |
| **MedicalNet-10** | **Aug** | 0.594 | **0.188** | 0.553 | 0.574 | **0.656** | 0.609 | **0.464** | **0.56** |
| MedicalNet-18 | w/o Aug | 0.621 | 0.313 | 0.579 | 0.6 | 0.533 | 0.131 | 0.392 | 0.463 |
| **MedicalNet-18** | **Aug** | 0.594 | **0.248** | **0.473** | 0.534 | **0.609** | 0.609 | **0.405** | **0.507** |
| MedicalNet-34 | w/o Aug | 0.621 | 0.309 | 0.588 | 0.605 | 0.567 | 0.139 | 0.45 | 0.509 |
| **MedicalNet-34** | **Aug** | 0.609 | **0.478** | 0.585 | 0.597 | **0.656** | 0.422 | **0.55** | **0.603** |
| MedicalNet-50 | w/o Aug | 0.672 | 0.568 | 0.696 | 0.684 | 0.45 | 0.625 | 0.516 | 0.483 |
| **MedicalNet-50** | **Aug** | 0.625 | **0.422** | 0.556 | 0.591 | **0.641** | **0.544** | 0.439 | **0.54** |
| 3D ResNet-34 | w/o Aug | 0.638 | 0.26 | 0.656 | 0.647 | 0.533 | 0.258 | 0.552 | 0.543 |
| **3D ResNet-34** | **Aug** | **0.655** | 0.271 | **0.661** | 0.563 | **0.617** | **0.24** | **0.6** | **0.609** |
| 3D DenseNet-121 | w/o Aug | 0.534 | 0.23 | 0.466 | 0.5 | 0.583 | 0.225 | 0.417 | 0.5 |
| **3D DenseNet-121** | **Aug** | **0.603** | 0.239 | **0.592** | **0.598** | 0.533 | 0.237 | **0.472** | **0.503** |
| Knowledge4D | w/o Aug | 0.603 | 0.263 | 0.621 | 0.612 | 0.517 | 0.245 | 0.506 | 0.511 |
| **Knowledge4D** | **Aug** | **0.638** | **0.261** | **0.67** | **0.615** | **0.567** | 0.246 | **0.565** | **0.566** |
| I3D | w/o Aug | 0.586 | 0.232 | 0.544 | 0.565 | 0.6 | 0.227 | 0.486 | 0.543 |
| **I3D** | **Aug** | 0.552 | 0.258 | **0.586** | **0.569** | 0.5 | 0.254 | **0.574** | 0.537 |
| FCNlinksCNN | w/o Aug | 0.586 | 0.236 | 0.563 | 0.575 | 0.5 | 0.239 | 0.449 | 0.474 |
| **FCNlinksCNN** | **Aug** | **0.586** | **0.164** | 0.53 | 0.558 | **0.567** | **0.171** | 0.428 | **0.497** |
| COVID-ViT | w/o Aug | 0.466 | 0.25 | 0.534 | 0.5 | 0.417 | 0.245 | 0.583 | 0.5 |
| **COVID-ViT** | **Aug** | **0.5** | 0.252 | **0.56** | **0.53** | **0.55** | 0.251 | **0.61** | **0.58** |
| Uni4Eye | w/o Aug | 0.603 | 0.247 | 0.607 | 0.605 | 0.55 | 0.237 | 0.576 | 0.563 |
| **Uni4Eye** | **Aug** | 0.586 | 0.265 | 0.592 | 0.589 | **0.583** | 0.277 | **0.645** | **0.614** |
| ADAPT | w/o Aug | 0.641 | 0.162 | 0.528 | 0.584 | 0.656 | 0.198 | 0.574 | 0.623 |
| **ADAPT** | **Aug** | **0.688** | 0.171 | **0.592** | **0.64** | **0.688** | 0.223 | **0.656** | **0.672** |

Table 7: Comparison of accuracy, brier score, specificity score and ROC-AUC score various 3D CNN-based and transformer-based models on tumor dataset. The numerical numbers of models with morphology augmentation are bolded when getting better performance.

set, and 25 subtype *0* and 34 subtype *1* in test set. We also applied the torchio augmentation to the training set and employed the zero-mean unit-variance method to normalize the intensity of all voxels within the images.

## B.2 Experiment Results

Following the experiment methods of Alzheimer's disease diagnosis, we tried to apply morphology augmentation to augment the atrophy of tumor brain mass, and compared the four metrics among various baselines and ADAPT. By analyzing the results in Table 7, we could see that ADAPT can achieve the best performance on various metrics. It outperforms other baseline models by 3.3%, 1.1% and 5.8% on accuracy, specificity and ROC-AUC score of test set. By comparing the performance between models with and without morphology augmentation, we found that except for I3D, all the ROC-AUC scores on test set were improved after applying the morphology augmentation. Especially for the FCNlinksCNN model and COVID-ViT model, without morphology augmentation, the model can't be trained successfully. These results show not only the superior of our proposed ADAPT, but also the necessity of the morphology augmentation method.

With the tumor dataset results, we proved our ADAPT can show its power in Alzheimer's diagnosis task, meanwhile can be expanded into other 3D disease diagnosis tasks, especially when the data type is brain MRI.

