# OpenReview forum: "ADAPT: Alzheimer's Diagnosis through Adaptive Profiling Transformers"
_ICLR.cc/2025/Conference — ICLR 2025 Conference Withdrawn Submission_

### Official Review · Reviewer_zRX8 · 2024-10-15

**Soundness:** 2
**Presentation:** 1
**Contribution:** 1
**Rating:** 1
**Confidence:** 4

**Summary:**

The authors propose a new transformer-based model for medical image classification with mutually-exclusive labels. They evaluate the proposed model (ADAPT) on Alzheimer’s Disease (AD) classification on four datasets (ADNI, AIBL, MIRIAD, and OASIS), and on glioblastoma classification (BRATS2021).
ADAPT is a 2.5D transformer (for non-medical readers: 2.5D approaches work on 2D slices in different orientations extracted from 3D volumetric images) that introduces four attention mechanisms, namely self-attention, dimension-specific attention, intra-dimension cross-attention, and inter-dimension cross-attention to leverage feature interaction across dimensions). Additionally, they propose morphology augmentation to increase and decrease atrophy of brain tissue, and an adaptive training strategy to let the model learn to extract discriminative slices from the volumetric images.
ADAPT outperforms 16 baselines on all tasks and saliency masks extracted from attention maps are presented.

**Strengths:**

- This work closely follows the pre-processing and data-splitting proposed by Wen et al. [1] to prevent data leakage in longitudinal data, which may result in overoptimistic performance measures when samples of one subject acquired at different time points are present in training and test sets.
- The model is compared to 16 baselines on both in-domain and out-of-domain test sets, which highlights its robustness in terms of out-of-domain performance.
- Increasing the number of samples by modulating pathological disease features and adaptively optimizing the model to focus on discriminative slices of 3D images are interesting ideas.
- The models are evaluated with many metrics, namely accuracy, Brier score, specificity, AUROC.
- The proposed adaptive training strategy poses an interesting and novel way of training the model to identify discriminative slices.

[1] Wen et al.: Convolutional neural networks for classification of alzheimer’s disease: Overview and reproducible evaluation. Medical Image Analysis, 2020.

**Weaknesses:**

Claiming SOTA performance:
The manuscript claims that ADAPT achieves SOTA performance for AD classification. However, Wen et al. [1] (cited in this work) reported balanced accuracies of 88% on very simple CNNs trained on ADNI compared to the 92% accuracy reported in the manuscript which does not take into account the large class-imbalance present in AD-related datasets like ADNI. Notably, 11 out of the 16 baselines achieves test accuracies less than a naïve classifier predicting the majority class on ADNI (total samples: 266 CN, 145 AD). This suggests that there may be opportunities to further optimize the baseline models.

Unclarities about the training of baselines and datasets:
It is unclear if the same training protocol (hyper-parameters, loss function for training with noisy labels, learning rate scheduler etc.) are used to train the baselines, which may explain the large discrepancies between baselines the proposed method.
It is stated in section 4.1 that details of the datasets can be found in Appendix A.2. Including comprehensive dataset statistics and known confounders (Age, Gender, mini-mental state examination, as discussed by Wen et al. [1] cited in the manuscript) would further strengthen the manuscript.

Morphology Augmentation:
The morphology augmentation proposed to increase dataset size is insufficiently explained, and Fig. 2 suggests that it only affects the ventricles. While current understanding of AD suggests that atrophy of AD-related brain regions leads to increased ventricle volume, this augmenting only the ventricle volume does not adequately model the complex atrophy patterns induced by AD. Next, randomly applying this augmentation strategy to MCI cases and assigning AD for increased and CN for decreased samples without further analysis of the individual samples poses the following problem: advanced (moderate) cases of MCI are likely to present severe (almost no) pathological changes other than ventricle volume in structural MRI. This approach may introduce significant noise into the dataset.
I recommend that the authors benchmark the morphology augmentation against other augmentation strategies like affine transformations or elastic deformations to better assess its effectiveness or elaborate on how the proposed morphology augmentation mitigates this issue.

Relevance
Classifying AD from CN is based on psychologically defined clinical stages [2].
If a patient progressed to the AD stage, structural changes in MRI are already strongly visible, but other (CSF) biomarkers change at earlier stages [2], which is why, the medical deep learning community has shifted to the more complex task of differentiating the three classes CN, MCI, and AD, differential diagnosis of dementia diseases, or perform time-to-disease prediction. I recommend evaluating ADAPT on the mentioned tasks to increase clinical relevance in a future submission.

Academic Standards:
The manuscript would benefit from revisions to meet the academic standards.
Specifically, the language lacks clarity, citep, and citet commands seem to be mixed up, claims are made without providing appropriate references, and sometimes seem to contradict the current medical knowledge on AD. Below are examples:
- “Also, 3D medical images are usually complex, making ViTs hard to pay attention to a special local feature that will play a crucial role in Alzheimer’s diagnosis. “
- “Many works have demonstrated remarkable achievements across various tasks, with several cutting-edge methods incorporating transformers for enhanced learning.”
- “U-Net and its variants dominate medical image analysis, which is widely used in image segmentation.“

- “A key characteristic of the AD-plagued brain is that, as the disease progresses, an increasing amount of brain mass will suffer from atropy. When this process is reflected in brain imaging, the there will be empty “holes” of the brain if one has AD.”
“Because clinicians usually diagnose AD using 2D slices but not the whole 3D MRI, we conjecture 2D slices may contain more valuable information.” This is an intriguing motivation, but overlooks the fact that humans inherently interpret visual data in two dimensions.

- “Alzheimer’s disease (AD) is a highly common neurodegenerative disorder that is usually diagnosed by structural alterations of the brain mass.” And “[…] physicians diagnosis alzheimer’s disease with MRI images […]”. AD is certainly not diagnosed using MRI only.

- “To overcome the vulnerability of misdiagnosis Despotovic ́ et al. (2015) […]”. The cited study compares medical segmentation methods.

[1] Wen et al.: Convolutional neural networks for classification of alzheimer’s disease: Overview and reproducible evaluation. Medical Image Analysis, 2020.

[2] Jack Jr. et al.: Tracking pathophysiological processes in Alzheimer’s disease: an updated hypothetical model of dynamic biomarkers. The Lancet Neurology, 2013.

**Questions:**

The saliency maps presented in Fig. 3 and Fig. 4 are very fine-grained and local. Typically, atrophy would appear uniformly across a brain region.
I recommend further evaluating saliency maps on a group level and regarding repeatability across model training.
Do the authors have any hypothesis on the fine-grained attention?

---

### Official Review · Reviewer_xvhq · 2024-10-30

**Soundness:** 2
**Presentation:** 2
**Contribution:** 2
**Rating:** 3
**Confidence:** 4

**Summary:**

This paper proposes a 2D method for AD classification that leverages 3D information from sMRI. The position information is included in the patch embedding to use the relative position of the 3D brain. Also, a morphology augmentation method is used to boost the classification performance.  Experimental results show that the proposed approach achieves better performance when compared with several methods.

**Strengths:**

1) The paper proposes a 2D method that can leverage information from 3D sMRI.
2) The position is leveraged into the model to capture the 3D information.
3) The proposed method achieves better classification performance for AD and NC classification task, when compared with several methods.

**Weaknesses:**

1) The classification between AD  and NC has limited practical significance in real-world applications.
2) The proposed used three views to learn the 3D information with a 2D method. But, this still results in a loss of significant 3D information.

**Questions:**

1) The three views and relative position information are used in the proposed methods. But, compared with the 3D method, the proposed method still loses the 3D image features.
2) Why not flatten the 3D patch to a sequence?  It seems more simple to handle the 3D image.
3) The comparison methods are quite outdated and should be compared with the latest methods.
4) The ACC in the validation set of ADNI is 0.9, but in the test sets of ADNI and AIBL, the proposed method achieves much better performance. How to explain these results?
5）Some methods perform particularly poorly. Was the experiment conducted fairly?
6）The classification of AD and NC lacks practical significance. The author should include MCI classification in the classification task.
7）Many 3D-based ViT methods have been proposed. What advantages does the proposed method have compared to them?

---

### Official Review · Reviewer_LBqg · 2024-10-31

**Soundness:** 2
**Presentation:** 1
**Contribution:** 2
**Rating:** 3
**Confidence:** 3

**Summary:**

The authors propose a task-specific Vision Transformer (ViT) model that integrates a novel attention mechanism, a specialized augmentation strategy, and an adaptive training approach to enhance Alzheimer’s Disease (AD) diagnosis. The performance of the proposed model is compared against a diverse array of methods using several AD datasets from various sources.

**Strengths:**

1. The authors present an interesting model design inspired by how physicians interpret MRI scans, which adds valuable context to the approach.
2. The attention maps generated by the model effectively highlight regions pertinent to AD diagnosis, demonstrating the method's interpretability for classification tasks.
3. The adaptive training strategy effectively guides the model to focus on more informative views and slices.
4. Algorithm 1 is well-structured, providing clarity on the complex setup and aiding reader comprehension.

**Weaknesses:**

1. The writing, particularly in the methods section, requires significant improvement. At least, clear and systematic definitions of all variables at the beginning of this section would greatly assist readers in following the paper.
2. Consistency in terminology will be helpful. The authors appear to use “view” and “dimensions” interchangeably, although they seem to refer to the same concept.
3. Figure 1 could better illustrate the main concepts and differences between IntraCAE and InterCAE, potentially better highlighting the concept of cross-slices attention and cross-views attention. The current presentation is somewhat confusing.
4. The role of the guide patch embedding is not sufficiently clear based on the description; additional explanations or relevant equations would enhance comprehension.
5. While the augmentation strategies for atrophy expansion and reduction are introduced, a clearer definition of the expansion or atrophy reduction element (bN) would be beneficial.
6. The comparison of numerous methods lacks focus. It would be more effective to highlight a few representative approaches with clear justifications for their selection, rather than overwhelming the reader with a comprehensive but uncurated list. For instance, results from different ResNet versions could be moved to supplementary materials.
7. The performance metrics for several compared methods in Table 1 are unexpectedly low (e.g., accuracy ranging from 0.4 to 0.65). Given the relative ease of the CN vs. AD task, this raises concerns about potential implementation issues or the need for better hyperparameter tuning. Based on the reviewer’s experience, a simple 3D CNN can achieve over 0.8 AUC on ADNI1 or 2 data. Further details regarding the implementations of the compared methods would clarify this.
8. Since CN vs. AD is a relatively straightforward classification task, it would be informative to evaluate the model's performance in more challenging scenarios, such as CN vs. MCI or CN vs. MCI vs. AD classifications.
9. The methodology appears to be specifically tailored for AD classification based on the employed augmentation strategies. In this venue, I would expect a more general method with broader applications or greater generalizability across different tasks. The inclusion of glioblastoma subtype diagnosis results in the appendix is a positive step, but it was not mentioned in the main manuscript. Also, similar to the previous point, several methods seem to perform worse than random guessing, which raises concerns about the implementation.
10. It would be helpful to know if cross-validation or random splitting was employed to assess the robustness and reproducibility of the model's performance. Including standard deviations or confidence intervals for the metrics would enhance the understanding of the results.

**Questions:**

Some mentioned above in weakness

---

### Official Review · Reviewer_PMEs · 2024-11-04

**Soundness:** 2
**Presentation:** 2
**Contribution:** 3
**Rating:** 5
**Confidence:** 4

**Summary:**

To balance the performance of 3D models with the computational efficiency of 2D methods, the proposed ADAPT framework extracts 2D slices from three orthogonal views of 3D AD MRI scans. These slices are then encoded using a carefully designed Transformer model that employs intra-view and inter-view cross-attention mechanisms to integrate information from the different views. Additionally, the number of slices selected is determined adaptively, and morphological augmentation is applied to expand or reduce ROIs which is crucial for accurate AD diagnosis. The proposed approach was trained on the ADNI dataset and tested on three external datasets, demonstrating its generalizability across diverse data sources.

**Strengths:**

- **The motivation behind this work is sound.**

By balancing the trade-offs between 3D and 2D models, ADAPT employs a 2.5D approach that achieves good performance with relatively few model parameters.

- **Both the model design and experimental setup are practical.**

The incorporation of morphological augmentation is particularly relevant, as it addresses atrophy—an essential characteristic of the AD-affected brain. Furthermore, training the model on the ADNI dataset and testing it on three external datasets closely mirrors real-world clinical scenarios, highlighting the model’s generalizability.

- **This work offers valuable insights into AD diagnosis.**

It emphasizes the importance of closely aligning model design with the clinical characteristics of AD. This will not only enhances the interpretability of the diagnostic process, but also makes it more applicable and insightful for AD diagnosis.

**Weaknesses:**

- **The comparison experiments could be expanded.**

The proposed ADAPT model is primarily compared with CNN-based models, along with a few Transformer-based methods, which may not provide a fully fair assessment. In my view, it would strengthen the study to include comparisons with baseline 2D/3D vanilla ViT and Swin-ViT models. Additionally, more SOTA models specifically designed for AD diagnosis, such as those in [1] and [2], should be considered, especially as some of these models have shown performance metrics that exceed those reported in this paper.

- **The writing and visual presentation could be further refined.**

The figure illustrating the ADAPT framework is challenging to interpret, as the QKV process of attention may not need to be displayed in full detail. Additionally, the workflow of components like SAE, DS-AE, IntraCAE, and InterCAE could be clarified for better readability. Some formula annotations are incomplete; for instance, the meaning of  $S_0$  in Eq.1 is not explained, and it is unclear if  $x_{guide}$  represents a layer or a feature. Providing complete and precise annotations would greatly improve the paper’s clarity.

*[1] Qiu, Zifeng, et al. "3D Multimodal Fusion Network with Disease-induced Joint Learning for Early Alzheimer’s Disease Diagnosis." IEEE Transactions on Medical Imaging (2024).*

*[2] Lei, Baiying, et al. "Hybrid federated learning with brain-region attention network for multi-center Alzheimer's disease detection." Pattern Recognition 153 (2024): 110423.*

**Questions:**

1. How are the regions for expansion or reduction determined in the morphological augmentation process? Is a segmentation algorithm required to guide this?
2. Since the slice selection is adaptive, what criteria are used to choose new slices from those not initially selected? For example, Line 381 mentions using 16 equidistant slices per view as a starting point, while Line 396 specifies 17 slices from the coronal and axial views. Could you clarify the approach for this adaptive selection?
3. I found the precision of the attention maps impressive. Did you employ any techniques, such as masking, to enhance their clarity or significance?
4. Was any data preprocessing applied to the images prior to training?
5. How would you explain the generalizability of your method across multiple datasets, given that no domain adaptation or multi-site loss appears to be incorporated?
6. Given that morphological augmentation appears to be a broadly applicable and reasonable approach to enhance model performance, how would you account for its failure to improve some of the compared models?

---

### Note · Authors · 2024-11-15

I have read and agree with the venue's withdrawal policy on behalf of myself and my co-authors.